



# Rainbows and Climate Change: A tutorial on climate model diagnostics and parameterization

Andrew Gettelman[1,2]

[1]National Center for Atmospheric Research, Boulder, CO, USA
[2]Now at: Pacific Northwest National Laboratory, Richland, WA, USA

**Correspondence:** Andrew Gettelman (andrew.gettelman@pnnl.gov)

**Abstract.** Earth System Models (ESMs) must represent processes below the grid scale of a model using representations (parameterizations) of physical and chemical processes. As a tutorial exercise to understand diagnostics and parameterization, this work presents a representation of rainbows for an ESM: the Community Earth System Model version 2 (CESM2). Using the 'state' of the model, basic physical laws, and some assumptions, we generate a representation of this unique optical phenomena as a diagnostic output. Rainbow occurrence and it's possible changes are related to cloud occurrence and rain formation which are critical uncertainties for climate change prediction. The work highlights issues which are typical of many diagnostics parameterizations such as assumptions, uncertain parameters and the difficulty of evaluation against uncertain observations. Results agree qualitatively with limited available global 'observations' of Rainbows. Rainbows are seen in expected locations in the sub-tropics over the ocean where broken clouds and frequent precipitation occurs. The diurnal peak is in the morning over ocean and in the evening over land. The representation of rainbows is found to be quantitatively sensitive to the assumed amount of cloudiness and the amount of stratiform rain. Rainbows are projected to have decreased, mostly in the Northern Hemisphere, due to aerosol pollution effects increasing cloud coverage since 1850. In the future, continued climate change is projected to decrease cloud cover, associated with a positive cloud feedback. As a result the rainbow diagnostic projects that rainbows will increase in the future, with the largest changes at mid-latitudes. The diagnostic may be useful for assessing cloud parameterizations, and is an exercise in how to build and test parameterizations of atmospheric phenomena.

## 1 Introduction

Parameterizations are simplified representations of natural phenomena used in climate or Earth System Models (ESMs). These simplified representations can be diagnostics (not affecting the model evolution), or prognostic (changing model evolution). A major source of uncertainty in models is representing critical earth system processes at the right scale (Hourdin et al., 2016). This work describes the representation of rainbows in an ESM.

Rainbows may seem trivial, but the basic conditions for a rainbow are particular relationships between clouds, rain and sunlight. Clouds (through cloud feedbacks) are the major uncertainty for climate feedbacks (Sherwood et al., 2020) and rain formation is critical for severe weather as well as understanding cloud adjustments to aerosols that modulate climate forcing (Bellouin et al., 2020). Ensuring that models simulate the right location and timing of clouds and rain is not trivial and may be





a useful integrated metric of the relationships in a model that may in fact be important for climate. The issues in representing rainbows are similar to other diagnostic parameterizations such as radar reflectivity (e.g. Fielding and Janisková (2020)). A rainbow diagnostic enables a unique analysis of the fidelity of model simulated phenomena to observations, and projections of how rainbows may change in the future, and why. The reasons can be related back to mechanisms of climate change.

Rainbows are also a tutorial exercise to understand how a diagnostic or parameterization is constructed. Rainbows require
applying basic physical laws to the state of the model, using some absolute quantities and some assumptions, to generate a diagnostic for when and where rainbows would be visible in a model. The diagnostic raises many general problems with parameterizations. Problems include the assumptions that are made, the difficulty of determining the validity at the right scale, and how to properly evaluate sub-grid scale processes in ESMs.

We start with a basic review of the essential physics of rainbows, previous scientific work on rainbows and a description of
the model to be used in Section 2. We then detail the implementation of the physics in the model in Section 3 and Appendix A. The representation of rainbows is diagnostic (it does not effect model evolution) but we will refer to the representation as a 'parameterization' or 'diagnostic' interchangeably. Section 4 contains a sensitivity analysis of the method, and then an overall evaluation of the representation based on current knowledge and available observations. We also look at the what the rainbow diagnostic can say about the impact of historical and future climate changes on rainbows. Section 5 discusses
critical uncertainties in the parameterization, as well as limits of the approach with respect to the limits of the parameterization. Section 6 summarizes the key findings and provides some overarching thoughts on how this development is relevant for many other more topical and 'important' representations in climate models.

## 2 Methodology

Rainbows are an optical phenomena. Businger (2021) and Haußmann (2016) both provide reviews of the basic physics, building
on work by Nussenzveig (1977). Haußmann (2016) discusses some of the unique physics and optics, and Businger (2021) also discusses the larger social context of rainbows with a focus on Hawaii. The scientific basis for rainbows is often attributed to Descartes (1637), as discussed in Werrett (2001). In addition, Diderot's Encyclopedia (Diderot and d'Alembert, 1751) lists an extensive entry for 'Arc-en-ciel', with earlier (and later) experiments discussed.

Rainbows occur due to the refraction of light by a raindrop, which separates the colors. Figure 1A illustrates a schematic
picture of refraction. Light passes through a raindrop and is scattered back to the observer following a fixed refraction angle of 42°. This requires a geometry where the light source is behind the observer facing the direction of the raindrop. The light source is usually the sun, which is what we will assume for this exercise, though 'moonbows' are also possible. Because of the refraction angle, a rainbow cannot be seen if the sun is higher than 42° from the horizon. Figure 1B and C illustrate that the size of a rainbow in the sky is inversely related to the height of the sun: a maximum (42° arc) when the sun is on the horizon
and disappearing when the sun is >42° above it. Figure 1B shows a large rainbow with a sun angle of 2° above the horizon (low sun, corresponding to a solar zenith angle from the zenith straight above of 88°. Figure 1C shows a small rainbow with a sun angle of 40° above the horizon (solar zenith angle of 50°).





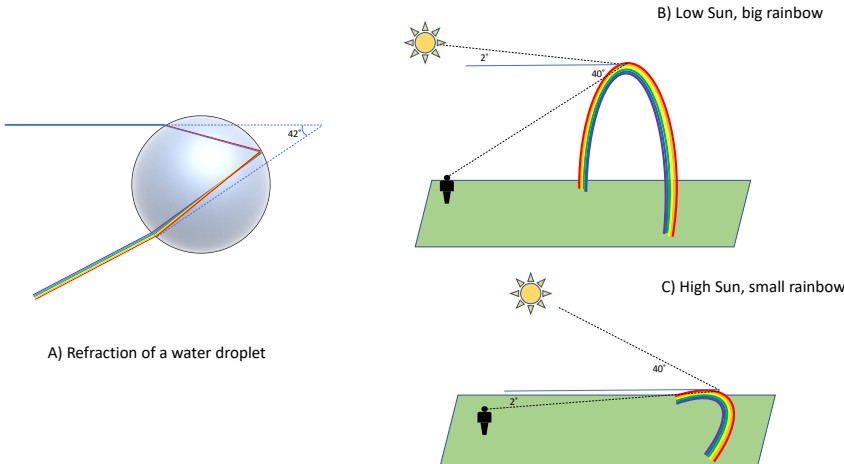

**Figure 1.** Schematic of rainbow optics. A) Refraction of a water drop and the size of a rainbow for B) Low sun angles and C) high sun angles. Similar to Businger (2021), Figure 3 and 5.

Businger (2021) and Haußmann (2016) provide many more details of the physics of rainbows, including double (secondary) rainbows (refraction twice within a rain drop) and many other interesting optical properties. For this first attempt, we confine

ourselves to the single 'primary' rainbow. Quantitative observations of rainbows are few and far between, as befits their status as an optical curiosity and not much more. However, Carlson et al. (2022) recently mined social media posts for rainbows and then used statistical methods to extrapolate to an atlas for 'rainbow days' that can be used for comparison. We will return to the potential for rainbow observations in section 5.

## 3 Implementation

Here we describe the modeling tool we will be using, and then a detailed description of how the representation of rainbows is constructed. Appendix A provides a more detailed description with a specific example as well as discussion of the issues involved in a generalized way to show how just about all representations of physical processes in large scale models face similar issues.

### 3.1 Model Description

The model we use is a state-of the art earth system model, the Community Earth System Model version 2 (CESM2) (Danabasoglu et al., 2020). The atmospheric component is the Community Atmosphere Model version 6 (CAM6) (Gettelman et al., 2019b), a General Circulation Model (GCM) with ∼100km horizontal and 500m-1km vertical resolution with 32 levels up to a top at 3hPa. The model has a hydrostatic dynamical core on a Cartesian (latitude-longitude) grid, and has a physical parameterization timestep of 30 minutes. Liquid cloud occurrence is estimated using the Cloud Layers Unified By Binormals (CLUBB)





unified turbulence scheme (Golaz et al., 2002), implemented in CAM6 by Bogenschutz et al. (2013). Ice clouds are treated as described by Gettelman et al. (2010). Cloud microphysics and precipitation formation is described by Gettelman et al. (2015b) and uses a bulk, two-moment representation of hydrometeors with prognostic two-moment rain and snow. Radiative transfer uses the Rapid Radiative Transfer Model for GCMs (RRTMG), described by Iacono et al. (2000).

The formulation of CAM6 implies important approximations we must consider when trying to represent rainbows. The

radiative transfer is plane-parallel with angular scattering, but the sun does not actually have a direction or location in the sky. We can calculate a solar zenith angle (SZA or $\theta_z$, the angle from straight up), based on location and time, and the solar radiation has the correct intensity and appropriate scattering, but not a specific direction. The sky is still blue because of this scattering. Clouds are divided into large-scale 'stratiform' clouds and deep convective clouds. Stratiform clouds are cubic volumes that fill a horizontal part of a grid volume at it's full depth, and randomly arrange themselves every 30 minutes. Stratiform clouds

have a distribution of cloud particle sizes, and a distribution of liquid and ice water mass, but not for purposes of radiation (they are uniformly gray). Stratiform clouds keep their shape but evolve every 15 minutes as water condenses and they produce precipitation. Deep convective motions ('thunderstorms') are basically small columns that move water up and down with simple microphysics that condenses water and detrains it out the top and precipitates it out the bottom. They disappear and re-appear again, at each time step. This is a strange and silent world (a hydrostatic model has no sound waves), but it does

provide a basis for representing rainbows.

### 3.2 Assembling a Rainbow

Rainbows need sun with a given angle and rain present at the same time. So to assemble rainbows we will step through a series of approximations. First, what are we trying to simulate? Since a rainbow needs to be seen to be observed, we are really looking at the 'potential' for rainbows given an observer is present. We do not incorporate the presence of an observer as Carlson et al.

(2022) did for rainbow 'hot spots'. We will estimate the frequency of time that a rainbow might be observed in a given volume, and the fractional coverage of that rainbow. Next, we assume that in a 100km horizontal grid box that all rain and rainbows occur in the same grid box. This assumption introduces some important limits on the diagnostic. Like many other 'column physics' parameterizations, we assume that we only need worry about the atmospheric state in a single column, not adjacent volumes. This means our diagnostic will break down at a scale in which the rain is not in the same grid box as the rainbow

and the observer. A rough estimate of such a scale is about 5–10km. At finer scales the parameterization would require some adjustment and communication across columns, which we will discuss at the end. This brings up the important point that even for a very simple parameterization or diagnostic the issues of scale and validity across scales must be considered explicitly.

The development and detailed illustrations of each step of the parameterization are described in Appendix A. First, we need to limit the sun angle ($\theta_Z$) to be within $42°$ of the horizon. We also need to know how much of the sky a rainbow could cover.

This again comes from geometry. A maximum rainbow size will occur when the sun is on the horizon ($\theta_Z = 90°$), and it will occupy a hemispheric cap of the sky of $42°$, diminishing to zero when $\theta_Z = 48°$. Second, there must be some clear sky in a grid box, so we have to set a maximum cloud fraction ($cmax$) that includes both convective ($A_c$) and stratiform ($A_{sr}$) cloud fraction. Third, near the surface of the earth below some level (which we define with a minimum pressure $pmin$), there must be





some precipitation in the grid box so we need to set a minimum for the stratiform ($rmin$) and convective ($rcmin$) precipitation. Convective precipitation is a surface rain rate ($P_c$), while stratiform precipitation has mass mixing ratio values ($q_{rs}$) throughout the column, so different thresholds are necessary.

Given these 4 parameters: $cmax$, $pmin$, $rmin$ and $rcmin$, a rainbow exists in a volume when:

1. $90° > \theta_Z > 48°$

2. $\max(A_c, A_{sr}) < cmax$

3. $(P_c > rcmin)$ or $(max(q_{rs}) > rmin)$

Where $A_c$, $A_{sr}$ and $q_{rs}$ are the maximum over model levels from the surface to $pmin$.

The 3 criteria above are applied with an appropriate selection of 4 parameters ($pmin, rmin, rcmin, cmax$). The parameter values chosen are the 'Default' values shown in Table 1. Note that these were not just picked randomly, but were the result of an initial assessment (sometimes called an expert elucidation, or more commonly an educated guess), with subsequent adjustment based on a more rigorous sensitivity analysis (see section 4.2). The results for a particular simulated time (Jan 8, 12 UTC) are illustrated in Figure 2. Figure 2A indicates where the criteria above are satisfied. For instantaneous data, rainbow frequency is binary (0 or 1) so that the time average is a true frequency of occurrence.

Rainbows of course do not fill the sky, and the probability seeing a rainbow may be more proportional to the area of the sky covered by a rainbow. As detailed in Appendix A, we define the fractional area of a rainbow $Frac_{RB}$ is the fraction of the hemisphere for a spherical cap of angle $\theta_Z$-48°, multiplied by the rain fraction ($A_r$):

$Frac_{RB} = 0$ for $\theta_Z > 48°$

$Frac_{RB} = ((1 - cos(\theta_Z - 48°))/2) \times A_r$ for $48° > \theta_Z > 90°$

$Frac_{RB} = 0$ for $\theta_Z > 90°$

Where the rain fraction $A_r$ is given by the maximum of the stratiform rain fraction ($A_{sr}$) and the convective cloud fraction ($A_c$) in the lower atmospheric layer defined by $pmin$:

$A_r = \max(A_{sr}, A_c)$

Figure 2B illustrates the rainbow fraction (frequency as zero or 1 multiplied by fractional area) for Jan 8, 12 UTC. Rainbows are found in an arc following the solar zenith angle, with fraction of the sky covered controlled by the solar angle $\theta_Z$ (larger with the sun near the horizon), and the fractional occurrence of rain (more rain area = larger rainbow)

## 4 Evaluation/Results

The rainbow diagnostic is put into the CAM specific interface for the cloud microphysics (Gettelman et al., 2015a). Eventually it will be developed as a stand alone code with an interface for the Common Community Physics Package (CCPP: https://github.com/NCAR/ccpp-scm). First we will illustrate basic climatology of where and when rainbows are expected to form. This will include a look at the diurnal cycle of rainbows in different regions, which is important to understand and provides



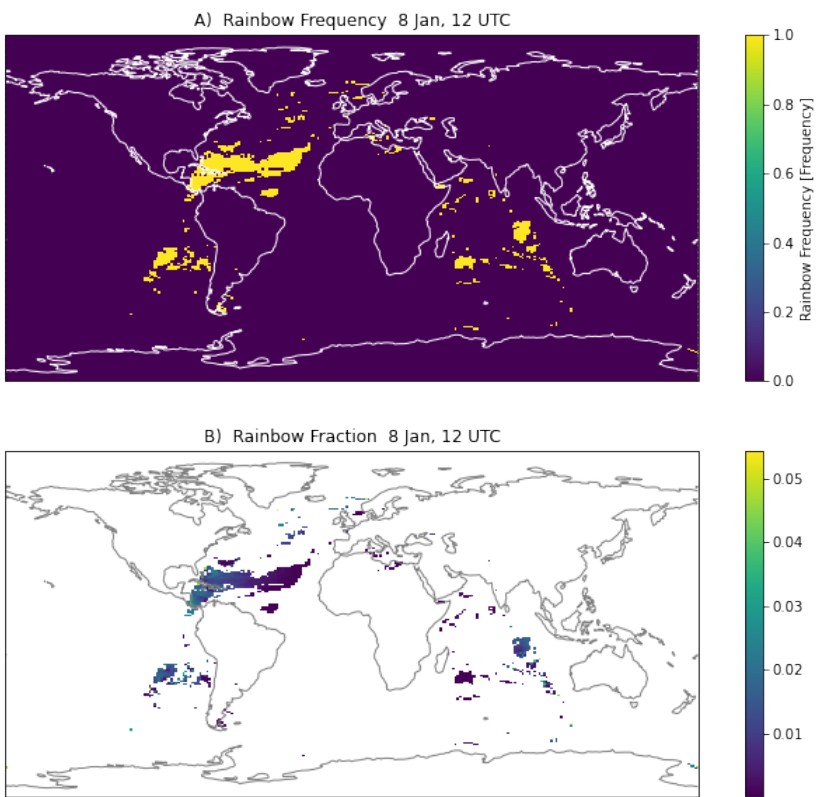

**Figure 2.** Instantaneous values on January 8th at 12 UTC. A) Rainbow Occurrence Frequency (1 = Rainbow). B) Rainbow Fraction of Sky covered.

**Table 1.** Rainbow Parameterization Parameter Values

| Param | Name | Units | Default | Range |
|-------|------|-------|---------|-------|
| pmin | minimum pressure for surface region | hPa | 850 | 950–800 |
| cmax | maximum cloud cover | fraction | 0.5 | 0.1–0.75 |
| rmin | minimum precip mixing ratio | $10^{-6}$ kg/kg | 1.0 | 0.01-10 |
| rcmin | minimum convective rain rate | mm/day | 5.0 | 0.01-10 |

140  insights on model fidelity. Then we explore the sensitivity of the parameterization to the four parameters. Four-year long simulations with climatological boundary conditions for the year 2000 are analyzed for these simulations. To analyze the diurnal cycle, short simulations were conducted with high frequency output for analysis.





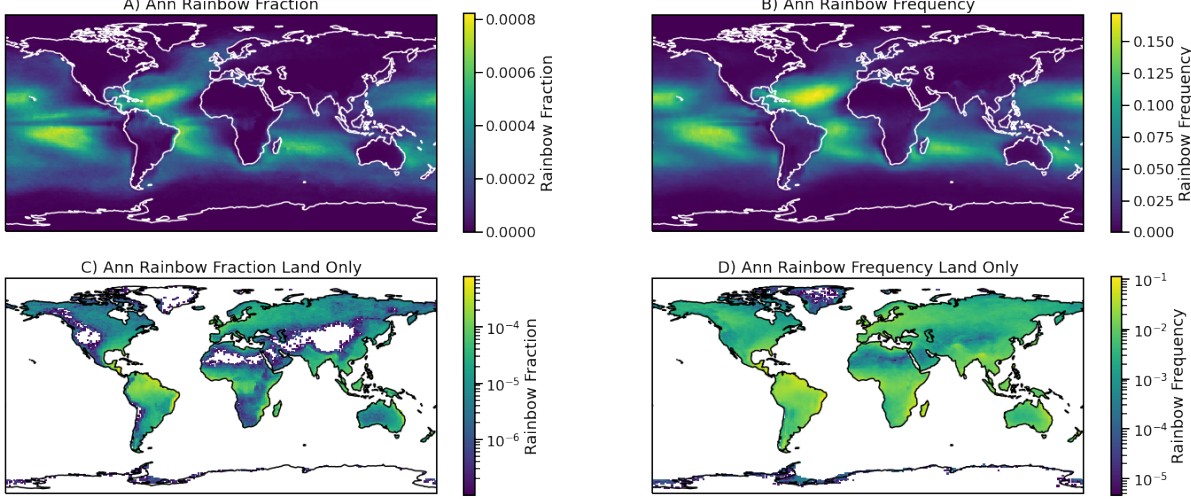

**Figure 3.** Climatological annual average (mean of 4 years) of diagnosed rainbow (A,C) fraction and (B,D) frequency over all locations (A,B) and just over land (C,D). Note the different scales: there is lower frequency over land.

Quantitative data on rainbow occurrence are scarce. Businger (2021) discuss qualitatively that rainbows are frequent over the Hawaiian islands due to island effects driving precipitation in the sub-tropical broken cloud regime. These regions have a diurnal cycle with rainbows morning and evening, with anecdotal and ethnographic evidence: native Hawaiian languages have many words for rainbows. Over mid-latitude land, broken clouds associated with thunderstorms in the afternoon and evening also produce rainbows. Similar situations permit rainbows in the evening hours in the summer in monsoon regions. But there do not seem to be quantitative climatologies of rainbows as would be desired to evaluate a parameterization. And of course being a ground-based optical phenomena, they are not observed from space, or even from aircraft (other 'bows' are seen, e.g. cloud bows, (Businger, 2021)). Carlson et al. (2022) have attempted to develop a metric for 'rainbow days' based on limited social media observations of rainbows 'trained' with precipitation and cloud data to project to other locations. Such data itself is difficult to evaluate, but in the absence of other data, we can use it for comparison.

### 4.1 Annual/Seasonal Means

Figure 3 shows the annual distribution of diagnosed rainbow fraction (sky coverage) and frequency of occurrence (anywhere in the sky) for all points (Figure 3A,B) and with a different scale to just highlight points over land (Figure 3C,D). Note all panels have different scales to highlight key features. The peaks are over the sub-tropical oceans, in regions of strato-cumulus (broken) clouds. Peak frequency is in the Subtropical Atlantic, with secondary maxima in the Central Pacific (near Hawaii) and also south of the Equator. There is much lower frequency over land.

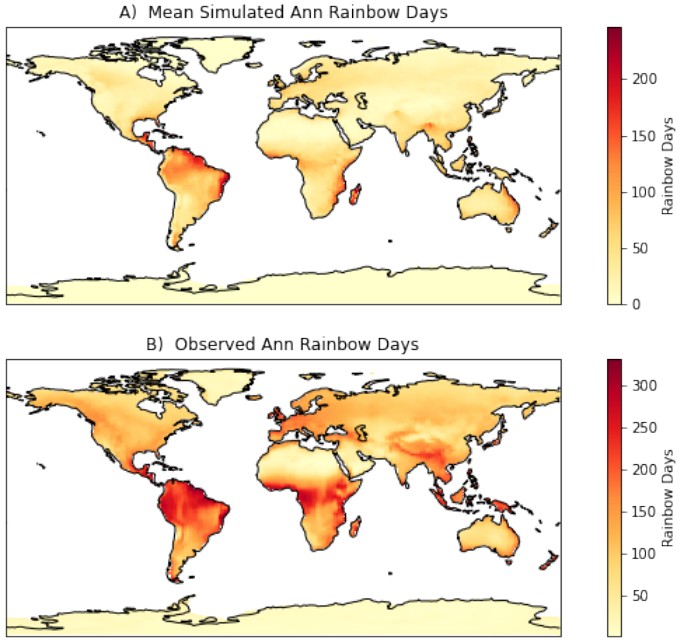

**Figure 4.** Annual average diagnosed rainbow days over land (see text) for: (A) simulations and (B) observations from Carlson et al. (2022).

For clarity, Figure 3C,D illustrate the same data, but only over land. Note the lower and logarithmic scales. Frequency over
land (Figure 3D), maximizes in the tropics and sub-tropics of the S. Hemisphere, with rainbow fraction (Figure 3C) highest in
the tropics. Coastal regions show higher values, including South and Central America and Africa. There are also coastal peaks
on the East coast of Continents (N. America, Asia and Australia).

To attempt to evaluate the parameterization quantitatively, Figure 4A is a map of 'rainbow days' to compare to Carlson et al.
(2022). Rainbow frequency for 4 years is turned into a binary value each day. Any rainbow frequency $> 0$ means that day is a
'rainbow day' at the given location. The number of rainbow days is calculated for each year and then an average over 4 years
is created. The variability (standard deviation) of rainbow days over each year at any point is about 10% of the value. The data
can then be compared to the data set from Carlson et al. (2022), derived by learning a relationship between rainbow observed
in social media images and precipitation and cloud fraction from reanalysis, illustrated in Figure 4B. The parameterization
locations and magnitudes are highly correlated (0.75) with the data set machine learned from observations, but are about 50%
lower (the slope of a point-by-point regression line is 0.47). As we will explore, this difference could be 'tuned' away if desired.
However, the 'observations' are subject to a number of potential biases, so perhaps further exploration of both the simulations
and observations are warranted.





Seasonally (Figure 5), rainbow frequency is found in the same regions over the ocean, with S. Hemisphere peaks in Sub-tropical Winter (JJA) and N. Hemisphere Peaks in Fall (SON) and Winter (DJF). This is interesting, especially the maxima in

the both hemispheres in sub-tropical in winter. The dominance in winter is due to lower sun angles expanding the times of day when a rainbow can form in these regions, while still sufficient precipitation occurs throughout the year. As noted by Businger (2021), the Hawaiian islands see solar angles that permit rain formation for 6.5h (58% of daylight hours) in summer but 8.5h (78% of daylight) in winter. Thus over the oceans in both hemispheres, the solar angles seem more important for the seasonal cycle than clouds or rain. Frequency and fraction track each other in location, and magnitude (note the slightly different scales

in each season to bring out the different locations). The pronounced banding of fraction (Figure 5 A, C, E and G) is an artifact of the discrete calculation of solar zenith angle every half hour time step (with 48 time steps per day).

The seasonal cycle over land (Figure A5) is different than ocean in many locations. There is little rainbow frequency over Northern Hemisphere land in winter and a peak frequency in spring over Europe and Asia. The tropics have relatively high frequency all year. The Southeast U.S. has moderately high frequency in all seasons but winter. S. Hemisphere land is generally

more sub-tropically situated, and thus has higher frequency in mid-latitudes, as well as high frequency in the tropics. S. Hemisphere arid regions seem to have higher rainbow frequency and fraction than the N. Hemisphere. This might be due to dry land-masses over the larger continents. More evaluation based on local conditions are warranted. In addition, we can analyze the diurnal cycle of rainbow diagnostics in key regions, including over land.

### 4.2 Diurnal Cycle and Sensitivity

One of the motivations for diagnosing rainbows is that a rainbow is an integrated metric of the representation of the diurnal cycle of clouds and rain in a GCM. GCMs have long standing biases in their representation of the diurnal cycle of precipitation, especially over land (e.g., Bogenschutz et al., 2018). This is driven largely by biases in the representation of deep convective systems (e.g., Xie et al., 2019). The expected diurnal cycle of rain is a strong diurnal cycle over land with afternoon and evening peaks, and a weaker diurnal cycle over ocean with a peak in early morning (Nesbitt and Zipser, 2003). We use the diurnal cycle

to test the sensitivity of the rainbow diagnostic to the choice of each of the four parameters in Table 1 ($rmin$, $rcmin$, $pmin$, $cmax$). The results in Section 4.1 above are 'default' settings.

To analyze the diurnal cycle and explore sensitivities, we archived timestep level output for two months: January and July. We took the model output for all the needed inputs to the rainbow diagnostic and ran it off-line. We also output the calculated rainbow diagnostic from the model and validated the off-line calculation produces the same answers as the in-line calculation.

The results are not bit-for-bit, due to slight differences in where the output of the model occurs in the time step loop relative to where the rainbow diagnostic is calculated, but results are qualitatively the same. Using the off-line output rainbow frequency and fraction is generated for a range of each of the four parameters separately in Table 1. To better visualize the sensitivity, monthly averaged diurnal cycles were created in 6 regions, 3 for January and 3 for July. Regions and their locations are listed in Table 2.

Figure 6 illustrates results for 3 regions in January, with sensitivity tests for the minimum rain mass for a rainbow ($rmin$). The base value used in Sections 3.2 and 4.1 is shown with a dotted line in all the figures (it will overlay one of the sensitivity





**Figure 5.** Seasonal average diagnosed rainbow fraction (A,C,E,G) and frequency (B,D,F,H), for different seasons. (A,B) December–February (DJF), (C,D) March–May (MAM), (E,F) June–August (JJA) and (G,H) September–November (SON).





**Table 2.** Regions Selected for Analysis of Rainbow Parameterization diurnal cycle sensitivity.

| Name | Season | Longitude | Latitude |
|---|---|---|---|
| Subtropical Atlantic | January | 280–320 | 10–30°N |
| Subtropical N. E. Pacific | January | 165–210 | 10–30°N |
| South America | January | 285–320 | 40°S–10°N |
| China | July | 75–120 | 22.5–45°N |
| India | July | 70–90 | 5–30°N |
| North America | July | 255–285 | 30–50°N |

tests, in this case $10^{-6}$ kg/kg). The plots are in local time, with a solid circle indicating daylight hours. Due to the averaging over a month, occasionally some rainbows will be seen beyond the circles. Generally, there is a pretty consistent and strong sensitivity of the rainbow fraction and frequency to the minimum rain mass needed for a rainbow for $rmin > 10^{-6}$, but not

for lower values. One main feature is that the diurnal cycle is unchanged, but the fraction and frequency are just scaled, though there are some differences in the relative peaks between morning and afternoon, as well as differences in sensitivity between regions, with less sensitivity over the land region of South America (Figure 6E and F).

Expected relationships between rainbows and the diurnal cycle are seen in different regions. Over the Subtropical Atlantic (Figure 6A) and Pacific (Figure 6C) oceans there are peaks in rainbow fraction in the morning and afternoon, with slightly

more rainbows seen in the morning, consistent with the diurnal cycle in oceanic rain peaking in the morning. However, over South America (Figure 6E and F) there is a much stronger afternoon peak in rainbow fraction and frequency, consistent with our understanding of the diurnal cycle of precipitation over land.

Results for $cmax$, the maximum allowed cloud fraction, in January show similar results: the same diurnal peaks by region, unchanged timing for different sensitivities, and strong sensitivity which we will analyze in July below (see Figure 7). Sensitiv-

ity of rainbow fraction and frequency to $rcmin$ and $pmin$ are small (not shown). This indicates that the minimum convective precipitation is not as important as the minimum stratiform precipitation in the model for rainbow formation, or that a wider range was chosen for the stratiform precipitation. It also indicates that the diagnostic is not sensitive to how deep a portion of the lower atmosphere is examined for rain and cloud. The $rcmin$ and $pmin$ sensitivity is also low in July, discussed next.

Figure 7 illustrates three different regions and their sensitivities to the maximum allowed cloud area ($cmax$) for July. Regions

chosen are over land in the Northern Hemisphere summer. Over mid-latitudes of China (Figure 7A and B) and North America (Figure 7E and F), there is a much more significant evening peak. Rainbow fraction peaks about an hour later (1800 v 1700 local time) over North America than China. Over India (Figure 7C and D) there is also a stronger afternoon peak, but higher relative frequency in the morning. The quantitative results are sensitive to the value of $cmax$, with almost an order of magnitude difference between rainbow occurrence (frequency or fraction) between $cmax$=0.1 and $cmax$=0.75. The value chose as default

($cmax$=0.5) was chosen based on anecdotal observations over North America indicating that rainbows can be seen with fairly significant low cloud coverage.



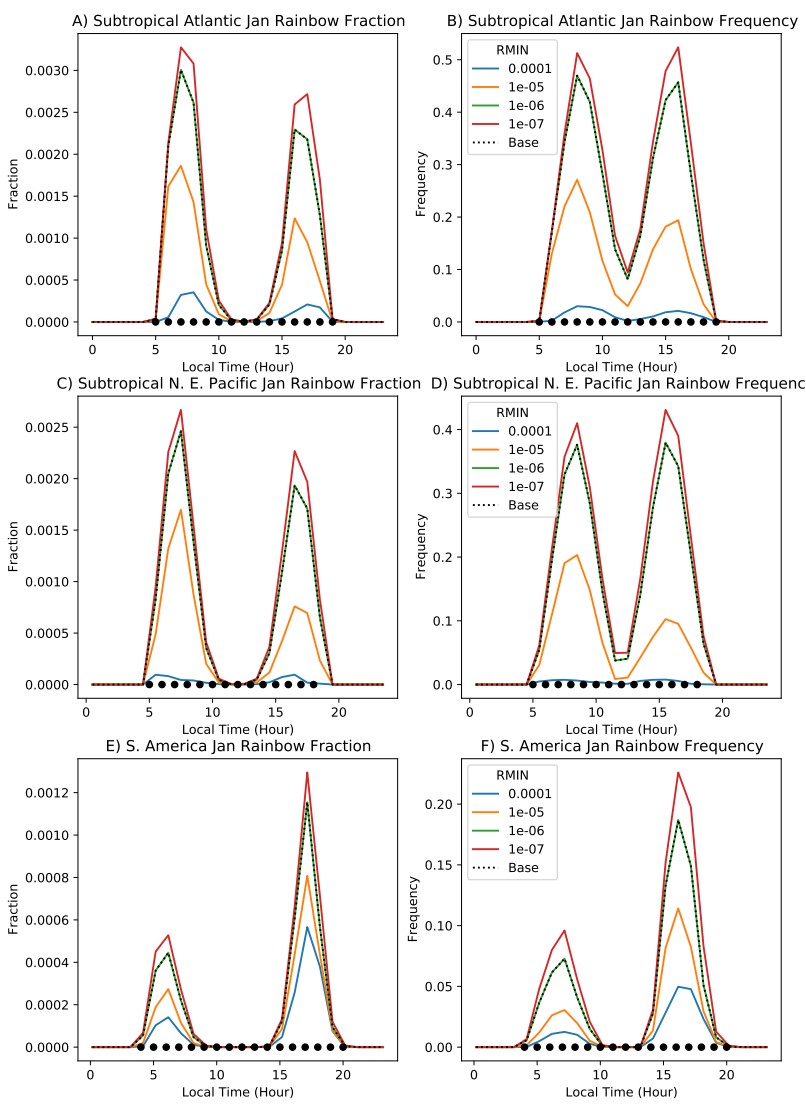

**Figure 6.** Diurnal cycle of rainbow fraction (A,C,E) and frequency (B,D,F) for 3 locations in January. Illustrated are calculations with different values of the minimum rain fraction necessary for a rainbow ($RMIN$). Different regions in different rows: Subtropical Atlantic (A,B- Top), N. E. Pacific (C,D-Middle) and South America (E,F-Bottom). Regions are indicated in Table 2.

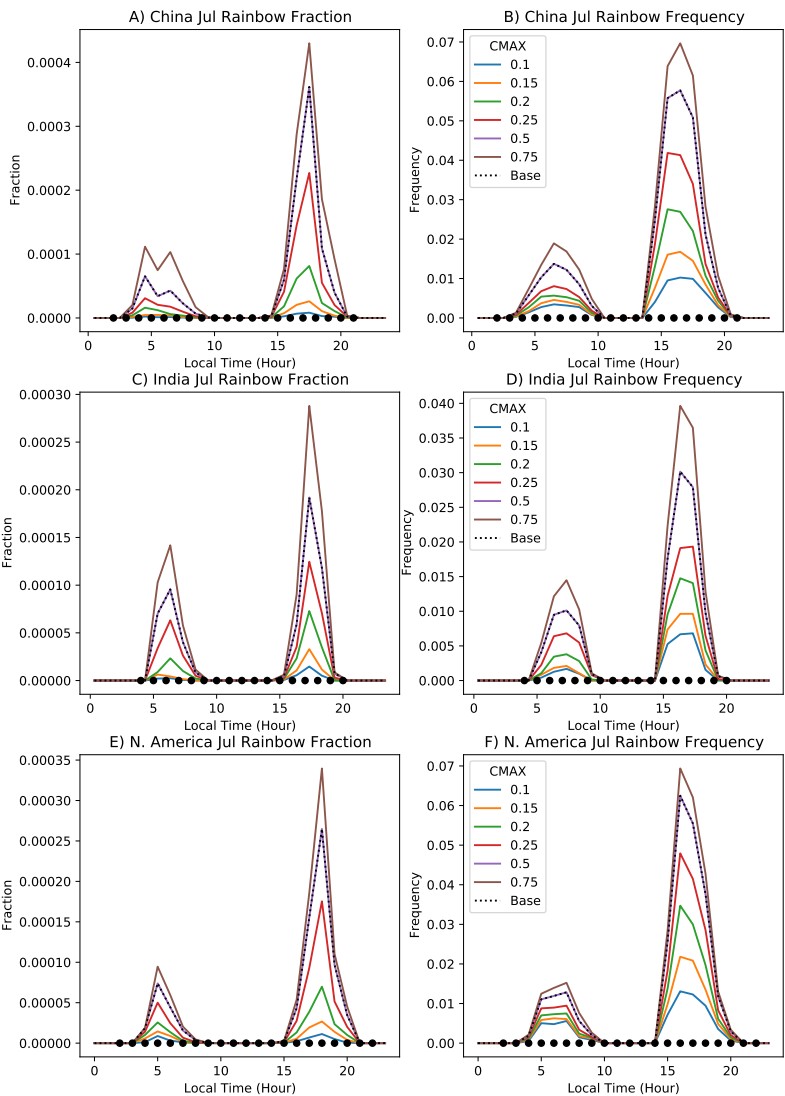

**Figure 7.** Diurnal cycle of rainbow fraction (A,C,E) and frequency (B,D,F) for 3 locations in July. Illustrated are calculations with different values of the maximum allowed cloud fraction (CMAX). Different regions in different rows: China (A,B- Top), India (C,D-Middle) and North America (E,F-Bottom). Regions are indicated in Table 2.





We have also performed some experimentation looking at maps of where rainbows occur with the different parameters. Results (not shown) indicate that the sensitivity tests quantitatively change the diagnosed frequency and fraction, but do not qualitatively change the location or relative magnitudes between different locations. Note that with respect to the observations

from Carlson et al. (2022) examined in Figure 4, the current parameterization produces on the 'high end' of rainbow frequency or fraction, and it could be adjusted by increasing the maximum allowed cloud fraction and lowering the rain threshold. This would increase the frequency of rainbows by 10–25% in each case.

### 4.3  Climate Change and Rainbows

Next we assess whether anthropogenic perturbations to clouds and climate will impact simulated rainbows. We set up identical

4 year global simulations but for two different configurations. First, identical to the control simulations, but with anthropogenic emissions of particulates (aerosol) and precursors set back to 1850 (Pre-Industrial or PI) conditions. This tests whether atmospheric pollution affecting clouds would have altered rainbow frequency or fraction. Second we can simulate climate change in an uncoupled (no dynamic ocean) atmosphere-land model by doubling the $CO_2$ concentration and uniformly increasing the Sea Surface Temperatures by 4°C (SST4K). The PI configuration is commonly used to examine anthropogenic aerosol effects on

climate, especially through changes to cloud properties (Bellouin et al., 2020). The SST4K configuration is a common method to assess climate responses and feedbacks in the atmosphere (Cess et al., 1989).

Since aerosols are the sites on which cloud drops form, more aerosols increase cloud drop number and brighten clouds (Twomey, 1977), with subsequent adjustments to cloud fraction, lifetime and/or water mass possible (Albrecht, 1989; Bellouin et al., 2020). Simulations with PI (1850) aerosols will have lower drop numbers and dimmer clouds (brighter in present day).

Figure 8 illustrates the absolute (top row) and percent (bottom row) change in rainbow fraction (left column) and frequency (right column) between the Present Day (PD) control run (as in Section 4.1) and a PI aerosol emissions simulation. Most of these changes if significant would be in the Northern Hemisphere. There are slightly higher changes in the Northern Hemisphere. The lower panel shows percent differences, with a threshold for when the rainbow frequency is greater than 0.01 and rainbow fraction is greater than 0.0002. These results indicate whether historical changes in clouds due to aerosol have affected

rainbows. Maximum decreases of the order of -20% are found in regions of high rainbow frequency (compare to Figure 3) over oceans. Changes over land are generally decreases. Note that these changes do not come from any direct scattering of sunlight due to increased particulates, which is not accounted for in the rainbow diagnostic. Also note that rainbow frequency changes are larger than differences in fraction (scale in Figure 8D are larger than Figure 8C).

Figure 9 provides an assessment of why the changes are occurring. The figure represents the difference in mean fields for

each of the inputs into the rainbow parameterization between the two runs in Figure 8 in the right column. We additionally take the pattern (Pearson) correlation between these differences and the rainbow fraction difference, as a gauge of what changes might be affecting the rainbow diagnostics. The pattern of differences is most closely related to changes in cloud fraction, with increases in cloud fraction contributing to a reduction in rainbows. Decreases in rainbows over N. Hemisphere land are associated with increases in cloud fraction. There is little change in the S. Hemisphere mid-latitudes as expected. This also





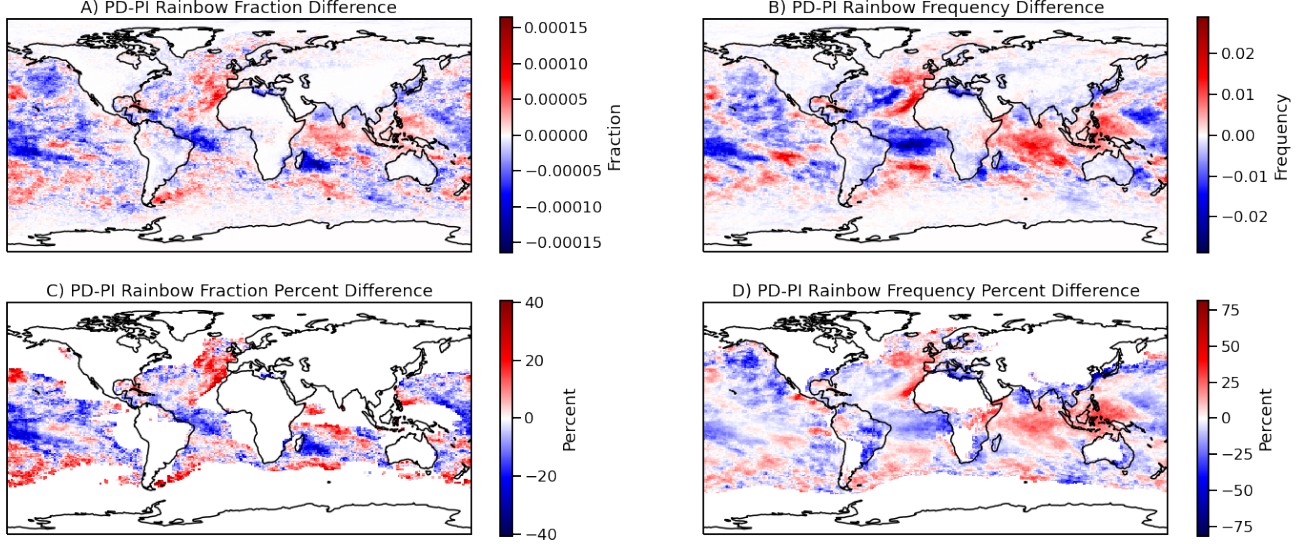

**Figure 8.** Annual mean absolute (A,B) and percent (C,D) difference in Rainbow Fraction (A,C) and Frequency (B,D) between present day (PI) and Pre-Industrial (PI, 1850) aerosol emissions. Percent differences only shown when PD rainbow frequency is greater than 0.01 and rainbow fraction is greater than 0.0002.

makes sense that the frequency changes are larger: frequency is a function of cloud fraction and rain, while fraction includes rain fraction (Section 3.2).

Finally, Figure 10 illustrates the impact of 'climate change' (increases in $CO_2$ and temperature) on the frequency of occurrence of rainbows. Rainbow fraction and frequency generally increase, with larger percent increases in rainbow frequency. Increases are small but consistent over land, and increases are large in the sub-tropics and into higher latitudes. Rainbow frac-

tion increases by 30–50%, with rainbow frequency in the subtropics and mid-latitudes over ocean nearly doubling in some locations. As indicated in Figure 9, this is mostly due to reductions in cloud cover as the planet warms (Figure 9A), with a pattern correlation coefficient in the tropics of -0.4. In addition, there are some tropical oceanic regions with decreasing rainbow frequency, and these regions have increasing cloud and rain fraction. Indeed, the correlation with maximum rain and rain frequency might be a correlation with cloudiness, and the correlations with rainbows a bit fortuitous.

The results are broadly consistent with changes in rainbows hypothesized by Carlson et al. (2022) by applying a machine learned rainbow model to future climate model output. Small increases in rainbow frequency were found over land and attributed to increases in rain and decreases in cloudiness. Some of the spatial patterns (increases over Indonesia, decreases over S. America) are also consistent. This is not unexpected as both methods rely on the same future data set (climate models).



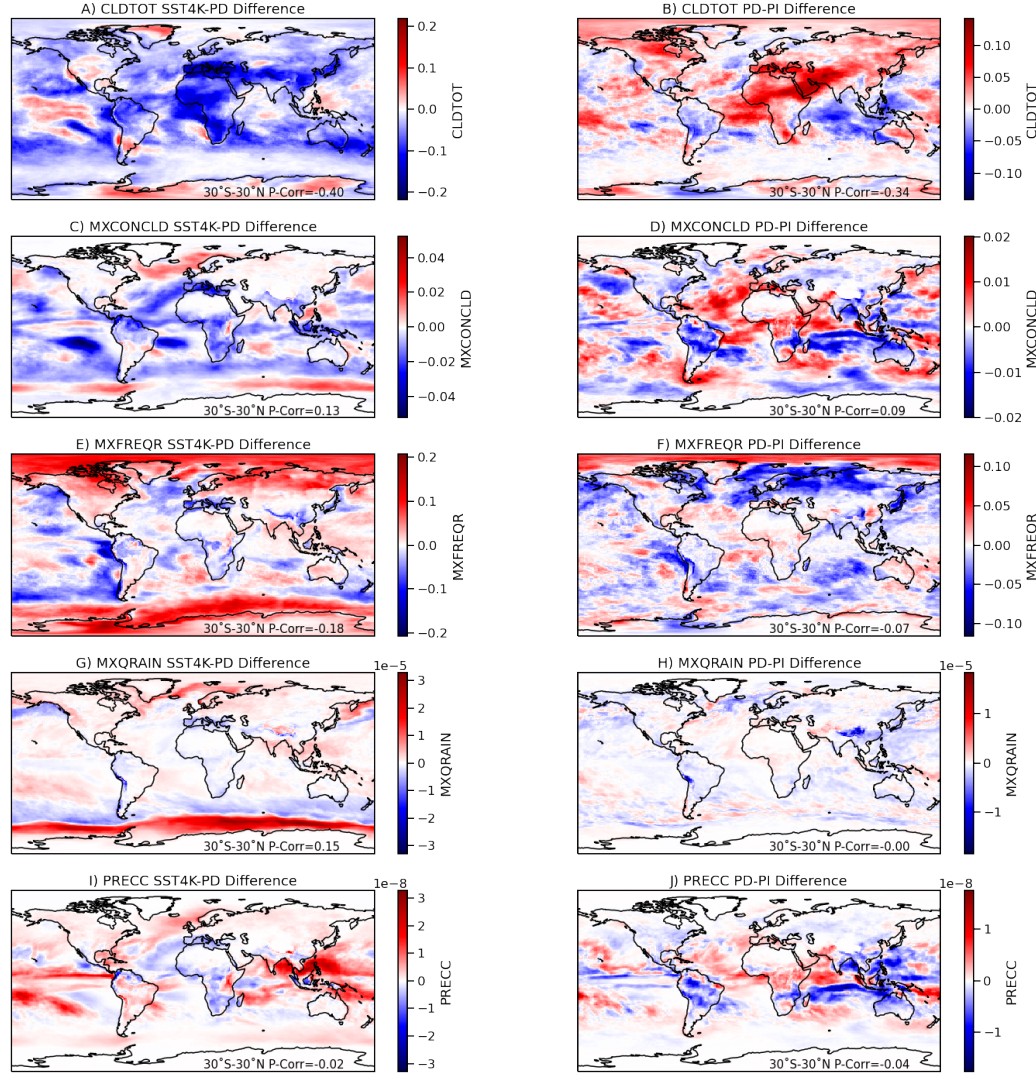

**Figure 9.** Annual mean absolute differences in the input fields to the rainbow parameterization for SST4K-Present Day (PD) (A,C,E,G,I: left column) and PI-PD (B,D,F,H,J: right column). Shown are (A,B) Total Cloud Cover (CLDTOT), (C,D) Maximum convective cloud cover (MXCONCLD), (E,F) Maximum Rain Fraction (MXFREQR), (G,H) Maximum Rain Mixing Ratio (MXQRAIN) and Convective Precipitation rate (PRECC).Numbers at the bottom indicate the Pearson pattern correlation coefficient between the differences in figures 8 and 10 averaged over 30°S–30°N).





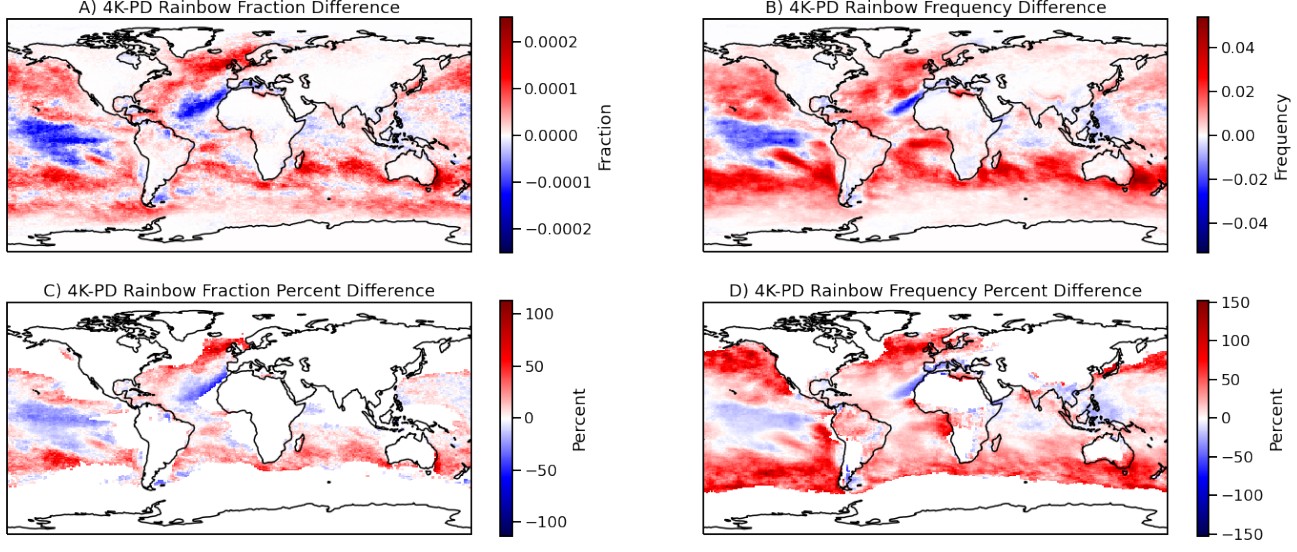

**Figure 10.** Annual mean absolute (A,B) and percent (C,D) difference in Rainbow Fraction (A,C) and Frequency (B,D) between 'climate change' simulations with $2xCO_2$ and SST+$4°K$ as described in the text. Percent differences only shown when PD rainbow frequency is greater than 0.01 and rainbow fraction is greater than 0.0002.

## 5 Discussion

The rainbow diagnostic parameterization represents most of the major intuitively expected features of where and when rainbows form. There is a strong constraint on the diagnostic from the basic physics which constrains the sun angle, and requires the presence of sun (partial cloudiness) and rain. This basic physics yields many of the resulting features of the regions, seasons and timing of the occurrence of rainbows. Simulated rainbow occurrence has broad fidelity to expected locations and diurnal cycle relationships. In regions with lots of precipitation throughout the year (like Hawaii), there are actually more rainbows

with lower sun angles than higher sun angles, but in other regions the seasonal cycle of precipitation dominates. There are some complex seasonal cycles in different regions that could be explored. The diagnostic is quantitatively sensitive to the assumptions for small amounts of stratiform rain and for the maximum cloud fraction permitted for a rainbow to exist, but it is not sensitive to convective rain and the lower layer height.

One of the most critical 'needs' is a better data set for evaluation of the rainbow diagnostic. We have made qualitative

comparisons with diurnal cycles over land and ocean, and attempted quantitative comparisons with a data set developed from machine learning the weather conditions for rainbows from limited observations. Comparisons are encouraging (particularly the pattern), with lower quantitative frequency than the observations. Adjustments could be made to better match the observations,





but this would require pushing the parameterization to the edge of the sensitivity range. This is a common conundrum for developing representations of phenomena. In this case the observations are likely highly uncertain (with possible selection bias), so we refrain from excessive tuning. Datasets or archives of all-sky camera imagery from key locations could be used to find rainbows with image processing, and calculate the frequency and fraction. This would have the advantage of being unbiased (continuous cameras) at a limited number of sites, and used to help calibrate other methods.

It is important to consider the scales of validity of the rainbow diagnostic. Many diagnostics or parameterizations make fundamental assumptions about the state of the atmosphere that are only valid at certain space and time scales. The diagnostic as developed for rainbows assumes that the rain, clouds and the observer are all in the same grid box of the atmosphere, and that no other information is needed from adjacent columns. The rainbow itself is seen by an observer but caused by rain in the same grid volume. This assumption is certainly valid for the 100km scales of a climate model analyzed here. But, given that CESM is now being run for resolutions down to 3km (Huang et al., 2022), it is important to consider where is the parameterization valid and can it be made to be scale selective (sometimes called scale aware). A rough estimate is that the rain and observer need to be in the same grid box as conceived here. But maybe that is not a problem as the potential rainbow would just be associated with a grid box that contains rain. The larger issue is that partial cloudiness is assumed, and for small scale models, clouds in a volume are either on or off. At that point, the parameterization would not work, and a non-column ('3D') treatment of rainbows would be necessary. This could be as simple as just looking for no cloud in the adjacent grid box in the direction of the sun, or as complex as using 3-D radiative transfer for ray tracing from the sun to rain and back to an observer.

## 6 Conclusions

A combination of basic physics, geometry and simple assumptions is effectively able to diagnose the potential for when and where rainbows would form in a GCM. The assumptions made in the development of the diagnostic parameterization and the different steps are similar to the way that other parameterizations are developed. The discussion starts with basic equations, and then off-line analysis with snapshots of model output. Then iterating over different parameter representations. Like many other formulations in a GCM, the parameterization has limits. The results agree very well qualitatively with limited available global observations derived from machine learning and social media imagery, with less frequency than 'observed'. Given selection bias in the observations, this may not be surprising. There is a need for more evaluation data, and it may need to be reformulated for models with small horizontal space scales, where the single column assumption breaks down. The diagnostic for rainbows is a good example of how complex phenomena can be represented using the existing state of a GCM, and the need for understanding the limits of those formulations.

Rainbows are not just an interesting optical phenomena, but provide important integrated metrics about key atmospheric processes. The representation of rainbows is found to be quantitatively sensitive to the assumed amount of cloudiness and the amount of stratiform rain. Neither affects the location or timing of the formation of a rainbow: only the potential frequency and fraction. Rainbows are seen in expected locations in the sub-tropics over the ocean where broken clouds and frequent





precipitation occurs. The diurnal peak is in the morning over ocean and in the evening over land. Sensitivity tests show little sensitivity to the diurnal structure or pattern of rainbows, mostly just to the quantitative fraction and frequency of occurrence.

This diagnostic enables analysis of simulations for aerosol forcing and climate change. Rainbows are projected to have decreased, mostly in the Northern Hemisphere, due to aerosol pollution effects increasing cloud coverage since Pre-Industrial times (1850). In the future, continued climate change forcing is projected to decrease cloud cover, associated with a positive

cloud feedback. The change in cloud cover is a general result for many climate models (Zelinka et al., 2020). Given that the rainbow diagnostic is sensitive to cloud cover (Figure 7, $cmax$), the rainbow diagnostic projects that rainbows will increase in the future, with the largest changes at mid-latitudes. These results seem consistent with sensitivity tests of the diagnostic. They are also consistent with use of the machine learning model by Carlson et al. (2022), which is not surprising since both projections are based on climate model output with similar future trends. More observations are needed to evaluate these

results, and refine the parameterization to be more quantitatively correct. A likely source of such data would be all-sky camera imagery with machine learning to find rainbow frequency and fraction at a series of stations. The limitation is that most of these locations would be over land, with a majority of rainbows found over the ocean. It would also be interesting to use this diagnostic in other earth system models to assist in evaluation of cloud and rain formation.

*Code and data availability.* Code described here is available in developmental versions of the Community Atmosphere Model (CAM), the

atmospheric component of the Community Earth System Model (CESM) as well as on zenodo at doi:10.5281/zenodo.7391777. A copy of key model outputs used in the analysis and the analysis code is available at the same location, doi:10.5281/zenodo.7391777

**Appendix A:  Rainbow Diagnostic Description**

Here we provide more detail and some examples of how the rainbow diagnostic was developed, with a case study using instantaneous fields for January 8, 12 UTC, as well as a description of the testing strategy and some supplemental plots of the

seasonal cycle.

**A1    Steps for Parameterization**

So what does it take to build a parameterization in a climate model? First is an understanding of the underlying physics. For rainbows, this is described in the text in Section 2. Second is the understanding of how the necessary physics is, or is NOT, described in a given model system. Then comes building a description of the physics consistent with the model system. All too

often this is done implicitly, without explicit understanding of the limits of the parameterization assumptions. We will try to make this explicit.

But the theoretical concept of parameterization belies an engineering aspect to development. The parameterization must be tested for algorithmic correctness (are the equations translated corrected), numerical stability and for the wide range of physical states found in a model. This applies whether the parameterization is theoretical (based on an equation) or empirical





(a fit to data). Note that empirical parameterizations can be as simple as a linear (or non-linear) regression, or as complex as a neural network, and the data can be either observations or another (often theoretical) model. See Gettelman et al. (2021) for an example of examples of this for the warm rain formation process, or Carlson et al. (2022) for how this was recently applied to rainbows.

Often this testing is done in a hierarchy of models, ranging from simple to complex (Jeevanjee et al., 2017). One often used
tool for physical processes in a model is the Single Column Model (SCM) where atmospheric motions are prescribed or forced and physical parameterizations are allowed to interact. SCMs range from single column energy balance models (Manabe and Wetherald, 1967), to complete representations of the processes in a general circulation model (e.g. Gettelman et al., 2019a). SCMs are used when interactions between processes are important. For diagnostic outputs that do not feedback on the model state (e.g. radar reflectivity or rainbows), often states of the model are used to run the parameterization off line (e.g., the Parallel
Offline Radiation Tool, PORT (Conley et al., 2013)). We use the off-line approach here.

At these various steps, there is often evaluation against observations for various outputs, ranging from direct comparisons (e.g. against radar reflectivity form observations) to indirect comparisons of emergent states of the climate system that result from parameterized processes (e.g. top of the atmosphere radiation budgets). Comparisons can use methods from simple statistical methods to complex emulation of data.

Finally, evaluations are often conducted to understand (and optimize) the sensitivity of resulting behavior from the parameterization against those observations. Sometimes this is called tuning (Hourdin et al., 2016). At the root of most parameterizations are uncertain 'parameters', whether an on/off threshold, the slope of a linear regression or the complex weights of a neural network. Sensitivity tests and optimization can be conducted again on a single feature of the climate state or system, or on a set of quantitative metrics. Section 4.2 explores the sensitivity of the scheme to the different parameters, and Section 4.1 illustrates
a comparison against the most relevant available data, as well as qualitative evaluation based on rainbow features.

The basic physics are detailed in the main text. Here we provide more illustration of the algorithm with an example snapshot.

The diagnostic parameterization was developed by outputing individual instantaneous time-slices of full 2D and 3D fields. The algorithm was first coded off-line in python to develop the logic. This took about 10 different iterations to design. Sensitivity tests of parameters were also coded off-line as described in the main text. Only then was the algorithm translated into
FORTRAN and implemented in the model, testing and debugging first in a single column framework (Gettelman et al., 2019b), and then in full simulations.

## A2    Description

First, we need to limit the sun angle ($\theta_Z$) to be within 42° of the horizon. In the model, $\theta_Z$ is measured from the zenith directly overhead, so at the horizon $\theta_Z$=90°. Figure A1A illustrates the distribution of $\theta_Z$ for January 8, 12 UTC, with lines marking
90° and 48°. Daylight at this time is when $\theta_Z < 90°$. The band when rainbows are possible due to the sun angle is between these lines ($48° < \theta_Z < 90°$). This is summer in the S. Hemisphere, with the sun centered overhead at about 25°S and 180°E. Rainbows could be found at any time of the day at 75°S, in morning and afternoon at 25°S, and throughout all the limited daylight hours from 25°N–65°N when the sun is never more than 42° above the horizon.

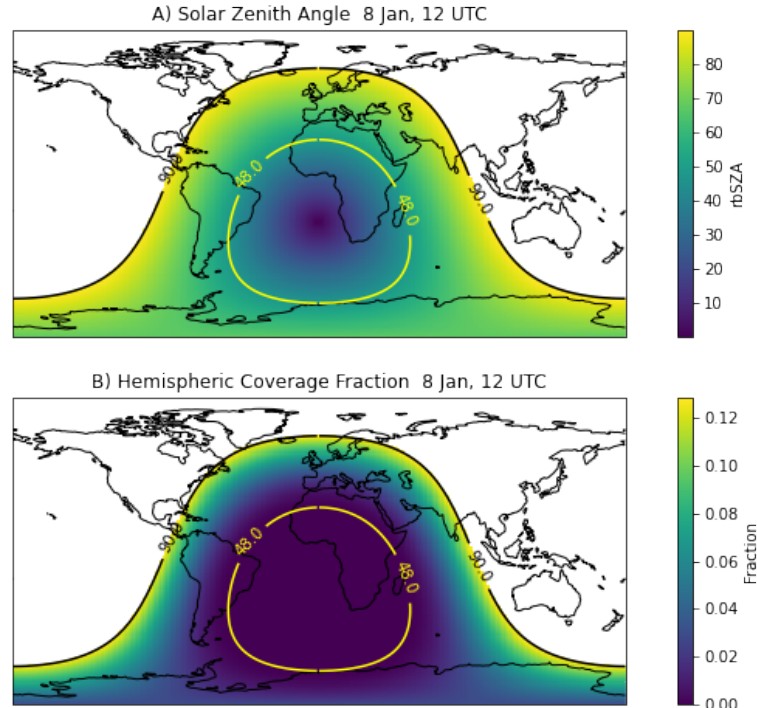

**Figure A1.** A) Solar Zenith Angle ($\theta_z$) on January 8th at 12 UTC. B) The hemispheric fractional coverage of a rainbow seen at the same date and time. Shown are the horizon of $\theta_z$=90° (black) and the minimum $\theta_z$ for rainbow visibility at the surface of $\theta_z$=48° (yellow).

Note that if we want to be able to scale rainbow fractional coverage by size (see below), we need to know how much of the sky a rainbow could cover. This again comes from geometry. A maximum rainbow size will occur when the sun is on the horizon ($\theta_Z = 90°$), and it will occupy a hemispheric cap of the sky of 42°, diminishing to zero when $\theta_Z = 48°$ (Figure 1 B and C). Solid angle theory allows the determination of the fractional sky covered by a spherical cap for angle $\phi$ as $(1 - cos(\phi))/2$, where here $\phi = \theta_Z$ -48° for $48° < \theta_Z < 90°$. This functional form is illustrated in Figure A1B, with a maximum fractional area of 0.13 at $\theta_Z$=90°.

Second, there must be some clear sky in the grid box, so we must set a maximum on the cloud fraction. Thus the maximum total cloud fraction of either convective ($A_c$) or stratiform ($A_s$) must be less than some value $cmax$. CAM6 provides this total cloud cover already maximally overlapped. This total cloud fraction for the same time as Figure A1 is illustrated in Figure A2. We pick an initial threshold of 0.5 for $cmax$ in Figure A2. Essentially the cloud cover is high over most of the planet (yellow),



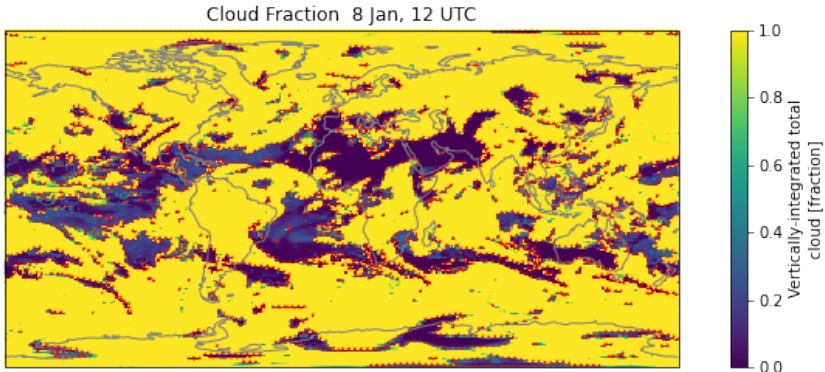

**Figure A2.** Total Cloud Fraction on January 8th at 12 UTC. Red dashed line is the 0.5 contour

with smaller totally and partially clear regions (green to dark blue). Only in the later regions is a rainbow possible with this
threshold.

Third, we require a minimum amount of precipitation in the grid box. For stratiform (large scale) precipitation ($q_{rs}$), which
is prognostic and present in each layer of the atmosphere, we set a minimum stratiform mixing ratio $rmin$. We pick a range of
pressures to sample to look for rain near the surface, selecting a minimum pressure $pmin$ for the top of the 'near-surface' layer.
For convective precipitation ($P_c$) we only have the surface flux, so we set a minimum convective rain rate $rcmin$. If either of
these is satisfied, then a rainbow is permitted. As an initial test we set $rmin$= 0.001 g/kg and $rcmin$= 5 mm/day with $pmin$=
850 hPa.

Figure A3 illustrates the different rain rates (Figure A3A,B) and the threshold criteria (Figure A3C). In Figure A3 A and B,
only rain above the threshold is shown. Since this is grid box averaged rain, the actual mass or surface rate is higher for partial
cloud cover. There is some non-zero stratiform rain over most of the oceans (Figure A3A), consistent with nearly consistent
light rain in CAM6, a common bias with many models (Stephens et al., 2010). Convective rain is found more concentrated
in the tropics (Figure A3B). If either of these thresholds are met, then the rain criteria is satisfied as true (=1 in Figure A3C),
which again occurs over most of the tropical oceans.

Given these 4 parameters: $cmax, pmin, rmin$ and $rcmin$, a rainbow exists in a volume when:

    1. $90° > \theta_Z > 48°$

2. $\max(A_c, A_{sr}) < cmax$

    3. $(P_c > rcmin)$ or $(q_{rs} > rmin)$

Where $A_c, A_{sr}$ and $q_{rs}$ are the maximum over model levels from the surface to $pmin$.

This defines the rainbow frequency of occurrence. As noted in the text, we also want to determine how much of the sky a
rainbow may occupy. To determine the fraction of sky coverage of a rainbow we use the spherical geometry of how much of a



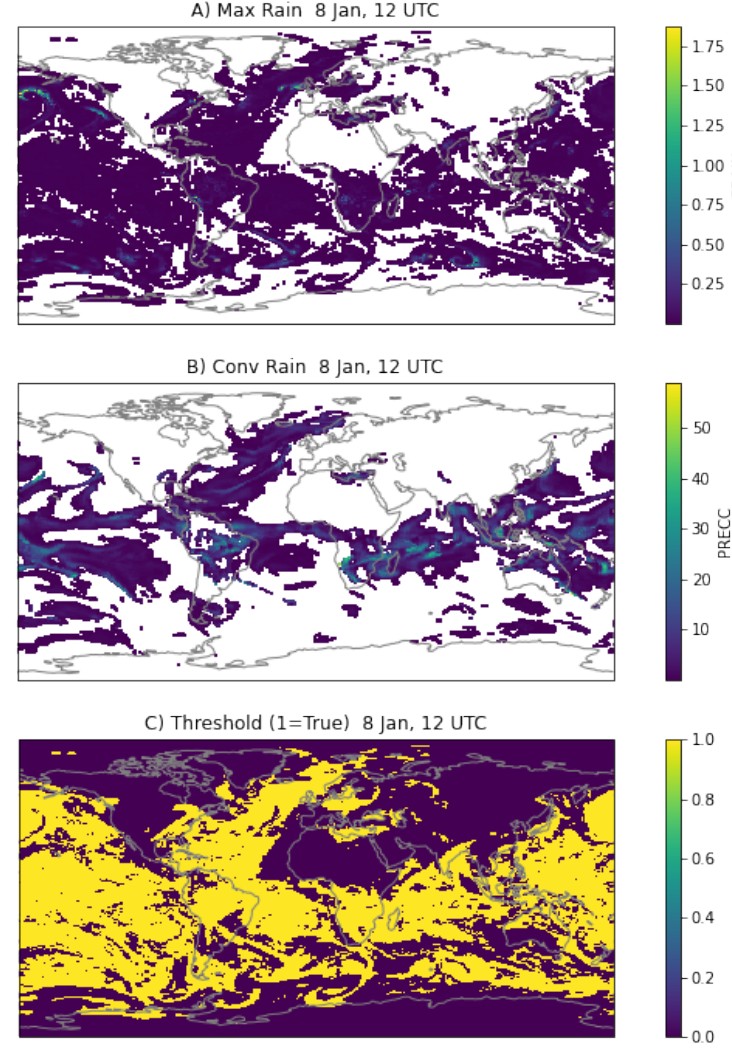

**Figure A3.** Instantaneous values on January 8th at 12 UTC. A) Maximum Stratiform Rain Mixing Ratio (g/kg), B) Convective Precipitation Rate (mm/day). Values only shown when larger than thresholds for (A) $rmin$ and (B) $rcmin$ as described in the text (otherwise white). C) Map of rain passing the threshold in yellow (either A or B have values).





hemisphere the rainbow could occupy, and a fractional occurrence of rain ($A_r$). $A_r$ is given by the maximum of the stratiform rain fraction ($A_{sr}$) and the convective cloud fraction ($A_c$) in the lower atmospheric layer defined by $pmin$:

$$A_r = \max(A_{sr}, A_c)$$

Figure A4 illustrates the maximum near surface stratiform rain area $\max(A_{sr})$ (Figure A4A), the maximum near surface convective cloud area, $\max(A_c)$ (Figure A4B), and the maximum total rain fraction $A_r$ (Figure A4C). Note that the stratiform

fraction is much higher, and the scales on Figure A4A and Figure A4B are different.

The fractional area of a rainbow $Frac_{RB}$ is the fraction of the hemisphere for a spherical cap of angle $\theta_Z$-48°, multiplied by the rain fraction ($A_r$) derived above:

$$Frac_{RB} = 0 \text{ for } \theta_Z > 48°$$

$$Frac_{RB} = ((1 - cos(\theta_Z - 48°))/2) \times A_r \text{ for } 48° > \theta_Z > 90°$$

$$Frac_{RB} = 0 \text{ for } \theta_Z > 90°$$

### A3   Development/Testing Strategy

The representation of rainbow frequency of occurrence ($Freq_{RB}$) and fractional coverage ($Frac_{RB}$) is purely diagnostic (it does not feed back on the model state). Thus the parameterization can be developed easily 'off-line'. CAM6 is run in a standard configuration (100km, 32 levels) for 1 month, with instantaneous output every timestep (30 minutes) for the required cloud,

rain and solar angle fields. The output is then used to analyze and test the parameterization off-line. The algorithm described above is not the only possible description of a rainbow, and other forms of the parameterization with more or less steps were tried. For example: the fraction of rain in a grid box could also be considered as a threshold value, but this involved assumptions about the convective rain area in the model. For this case, an 'Occam's Razor' approach is used to simplify the representation as much as possible and minimize the number of parameters.

Testing was conducted off-line, and then the resulting parameterization coded back into the model, in the interface to the cloud microphysics routine, where it will eventually become a small stand alone diagnostic code. The product is numbers: a fractional area coverage of a rainbow in each grid volume at each time ($Frac_{RB}$), and a frequency flag set to 1 if there is any rainbow, zero if not ($Freq_{RB}$). The average of that flag over a time period yields the frequency of occurrence. The advantage of this approach is that the model test run can be conducted again with the in-line calculations for $Frac_{RB}$ and $Freq_{RB}$, and

validated against the off-line estimate for debugging purposes. It also enables rapid off-line sensitivity tests to be developed. Table 1 lists the default parameter values and ranges selected for the sensitivity tests in the text.

### A4   Seasonal Cycle over Land

The seasonal cycle over land is illustrated in Figure A5 from 4 year climatological simulations. There is little rainbow frequency over Northern Hemisphere land in winter (DJF-Figure A5B) and a peak frequency in spring (MAM-Figure A5D), particularly

over Europe and Asia. The tropics have relatively high frequency all year. Fraction adds the sky coverage, and is higher for low sun angles (e.g., the discussion of a winter peak in Hawaii above). So the combination means rainbow potential fraction peaks in N. Hemisphere spring, including over Europe. The Southeast U.S. has moderately high frequency in all seasons but

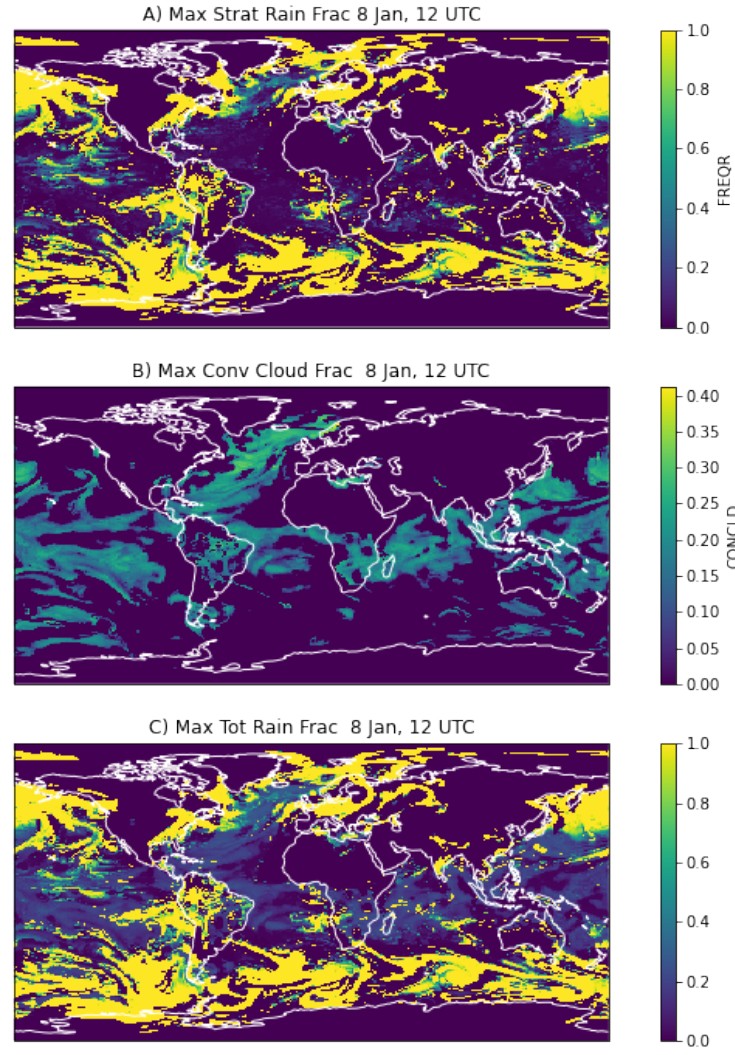

**Figure A4.** Instantaneous values on January 8th at 12 UTC. A) Stratifom Rain Fraction B) Maximum Convective Cloud Fraction and C) Total Rain Fraction as the maximally overlapped combination of A and B.



winter. S. Hemisphere land is generally more sub-tropically situated, and thus has higher frequency in mid-latitudes, as well as high frequency in the tropics. S. Hemisphere arid regions seem to have higher rainbow frequency and fraction than the N. Hemisphere. This might be due to dry land-masses over the larger continents. More evaluation based on local conditions are warranted. See discussion of observations below. In addition, we can analyze the diurnal cycle of rainbow diagnostics in key regions.



**Figure A5.** Seasonal average diagnosed land only rainbow fraction (A,C,E,G) and frequency (B,D,F,H), for different seasons. (A,B) December–February (DJF), (C,D) March–May (MAM), (E,F) June–August (JJA) and (G,H) September–November (SON).



*Author contributions.* AG wrote the code, designed the experiments, did the analysis and wrote the paper.

*Competing interests.* No competing interests are present

*Acknowledgements.* The National Center for Atmospheric Research is Sponsored by the United States National Science Foundation. The Pacific Northwest National Laboratory is operated by Battelle for he United States Department of Energy. Thanks to Kate-Thayer Calder for software engineering assistance. Thanks to Christine Shields and Po-Lun Ma for comments.



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
