# Peer review of "Rainbows and Climate Change: A tutorial on climate model diagnostics and parameterization"

_EGUsphere, 2023_

## Author Response (AR1)

We thank the reviewers for their kind words and the attention to detail in the reviews of the manuscript. We have made all the suggested corrections and to the manuscript suggested. We decided not to move around the figures from the appendix as suggested by one of the reviewers. Other than that we have agreed with and adjusted the manuscript to accommodate all the reviewers comments and concerns.

Review Comment #1:

This manuscript presents integration of rainbow representation into an Earth System Model (ESM) and develops an understanding of rainbow sensitivity to model parameters, rainbow spatial and temporal (seasonal and diurnal) variation, and rainbow alterations under global anthropogenic change. The author argues that this paper/approach can help teach how a diagnostic is constructed, and suggests that to better incorporate rainbows into ESMs, sampling rainbows using a strategic and consistent approach may be useful. The manuscript builds on recent publications focusing on the conditions under which rainbows can be seen and presenting a model built from empirical observations of rainbows from social media. Thus, the manuscript is timely and relevant. Since the manuscript focuses on model parameterization, it is appropriate for this manuscript type (methods for assessment of models), and I believe the concept – simulating rainbow occurrence in an ESM – is novel. My general and specific comments are described below.

GENERAL COMMENTS

Sensitivity of predictions to model parameter changes. The author examines model sensitivity to changing parameters across diurnal cycles. What is the sensitivity of predicted rainbow fraction/frequency to changes in model parameters under PI and SST4K simulations?

>> This is an interesting question. We re-ran sensitivity tests as suggested for the PI and SST4K cases. The sensitivities for all the parameters are qualitatively the same with only small quantitative differences when PI or SST4K conditions are analyzed. Noted in the text now.

Rainbows versus clouds/rain in diagnosing model performance. Part of the justification for including rainbows in an ESM is that it is difficult to ensure that models "simulate the right location and timing of clouds and rain". I'd love to hear more about why directly diagnosing whether the simulation of clouds/rain is accurate or not is difficult, and why a proxy (rainbows) that combines both rain and clouds might be easier or uniquely useful. After all, we have developed ways to observe clouds and rain, such observations can help test model performance, and good observations don't (as the author points out) yet exist for rainbows. Greater development of the concept that rainbows are a useful proxy for these critical earth system model features would strengthen the manuscript.

>> We have added some comments to the introduction to discuss this point.

*Extensive observation networks exist for clouds and rain from the surface and from space, ranging from rain gauges to surface radar to satellites. Often the presence of rain is hard to detect, either because it does not hit the ground and/or it may be too light (or in too small a region) to see from space. Rainbows can be a stark visual identification of rain that other measurements miss.*

Observations. It seems that not only rainbow observations, but rainbow observations coupled with excellent synchronous rainfall and cloud observations, would best refine the parameterization. Thoughts on how all relevant parameters might be measured would be welcome.

>> Another excellent point. We have expanded the discussion of observations in section 6 to encompass observations of clouds and rain, and how with rainbow observations these could be used to constrain the parameterization.

Pronoun. Consider using "I" rather than "we" for this single author paper.

>> Thanks for the comment. I'll let the editorial staff decide which is more appropriate. It seems a bit presumptuous to change 'we' to 'I'. I think 'we' sounds a bit more inclusive for the reader.  But will do whatever the editing staff think it is more appropriate stylistically. I guess I'm not used to writing single author papers!

SPECIFIC COMMENTS

P1L24. The fraction/amount of clouds and rain, in addition to timing and location, seems critical especially in light of the finding that the model is very sensitive to cloud and rain fraction

>> Agreed. Added 'frequency/fraction'. Also on line 123 deleted the word 'more'.

P2L29. A tutorial for whom – students learning how to use ESMs?

>> Clarified.

P2L32-33. This sentence (beginning with "The diagnostic") is quite vague to me and it seems like it could be better developed and/or supported with sources discussing these difficulties (admittedly, I'm not an expert in ESMs).

>> Reworded. Added a reference to a recent paper on parameterization and parameter uncertainty with more specifics, and merged with the following sentence which has more specifics.

P3L60-61. Carlson et al. (2022) argue that rainbows provide ecosystem services, making them more (potentially) than an optical curiosity.

>> Added a reference to this in the 2nd paragraph of the introduction

P4L100. How was this rough estimate of scale developed?

>> Rederived. Initially it was a guess, but thanks for this comment I've given it further thought. If we assume that a rainbow is in the lower 2km of the atmosphere (boundary layer), then refraction limits the height above the surface (within 42 deg of the horizon), taking half that size (21 deg) means that for a rainbow at 2km height it would be about 5 km away (2km/tan 21˚). It could be farther away but would be smaller.   Now noted in the text.

More broadly, it is interesting to think about the relativity inherent in viewing rainbows and how this translates to the model. The location of a rainbow is always relative to the location of a viewer (a viewer at a different location may not be able to see the rainbow that exists for the first viewer). This relativity means that modeling rainbows is different than modeling something that is made of matter we can touch, like a cloud or a rain droplet, that is present no matter where the observer is located. A more in depth discussion of this issue may be warranted.

>> It is already noted that the diagnostic is a 'potential' for rainbows, and now clarified a bit, and noted that the potential is the 'maximum likelihood' it is possible to see a rainbow under the given conditions if an observer were present in the right place in a grid volume at a given time.

P5L116. I am not an ES modeler myself and am not sure what rain fraction is as described above in Line 116 (the maximum over model levels?). Consider clarifying this definition if this is not clear to the expected reader of this journal.

>> Clarified. These are not fractions, they are actually masses of rain.

P5L117-120. It seems that a max cloud fraction of 0.5 is a reasonable guess in the absence of other good information. I do think it would be worth mentioning, however, that Carlson et al. (2022) used a maximum cloud fraction of 0.96 in their model. Eyeballing their figure 3, >50% of rainbow observations occurred with cloud fraction >0.5. Given that the model is sensitive to cloud fraction, I urge the author to discuss in greater detail why they chose a much lower value than the previous study.

>> Anecdotally rainbows with broken clouds get hard to see: in order to see the sun from below a cloud, the depth of the cloud will block rain as it gets too thin. Even with 50% cloud cover, there are few patches of sky visible, and the odds of the sun shining into them are low. I was surprised by the Carlson plot. Given that there is not a strong decrease in rainbows as cloud cover approaches 1,  it seems obvious that the reanalysis cloud cover data is subject to errors.  Now noted in the text.

P5L124. Please explain more clearly / in greater detail why rain fraction by the fraction of the hemisphere here in the main text (it is explained nicely in the appendix).

>> Added a sentence more about how it is defined and referenced Figure 1 (as in the appendix

Figure 2. Consider adding the panel from the appendix that shows where rainbows are possible based on sun angle here.

>> I experimented with this, but think it best to keep the entire discussion of the parameterization in the appendix, with a reference in the main text, and just the final results in the main text.

P8L159. This first sentence repeats the same concept in the previous paragraph (land, lower values).

>> Clarified (deleted last sentence of previous paragraph that was duplicative).

P9L174-175 "especially the maximum in the both hemispheres in sub-tropical in winter" has some apparent grammar issues.

>> Whoops. Clarified and merged with the following sentence for clarity.

P9L176. "while still sufficient precipitation" – I assume that this is liquid not overall precipitation, even in the winter?

>> Clarified that this is liquid precipitation (ice will not form rainbows, it has different refraction properties).

Maybe worth mentioning this difference in the phase of precipitation over land versus ocean and how this drives seasonal rainbow differences (if indeed there are differences in precipitation phase over land and ocean)

>> Good point. Noted.

P9L190-194. It seems that this argument/justification might be more compelling in the introduction of the manuscript.

>> Good point. moved as suggested.

P9L198-199. As a non-ES modeler, I found this language on outputs/inputs and offline/inline confusing and think it could be revised for clarity.

>> Clarified. Removed offline/inline and tried to think about more common phrasing.

P14L249. What is a lower drop number? Please clarify.

>> Clarified. "lower concentrations of cloud drops within clouds"

P26L456. "See discussion of observations below" seems like it might be modified to point to discussion in the main text?

>> Whoops. Clarified.

Works Cited

Carlson, K. M., C. Mora, J. Xu, R. O. Setter, M. Harangody, E. C. Franklin, M. B. Kantar, M. Lucas, Z. M. Menzo, D. Spirandelli, D. Schanzenbach, C. C. Warr, A. E. Wong & S. Businger (2022) Global rainbow distribution under current and future climates. *Global Environmental Change,* 77, 102604.

Review Comment #2

Review of "Rainbows and Climate Change"

Summary:

This is an interesting paper that summarizes the various important aspects and challenges of constructing a parameterization of a physical process. In this case, the parameterization is a diagnostic, not affecting the model solution, but the principles are the same. A "rainbow frequency parameterization/diagnostic" is described as way of illustrating how one would construct, validate, and tune any type of physics parameterization. This is a really cool idea and a very nice paper. I think this paper should be mandatory reading for any course/program on numerical modeling. Even for non-developers, the paper is very useful for anyone making use of atmospheric model output (any type, not just GCMs) because it gives a useful and easy-to-understand description of the challenges, limitations, etc. of a parameterization of a physical processes (or field). I made a few suggestions for fine-tuning, but basically I think this manuscript is in good shape and nearly ready for publication.

Jason Milbrandt

Minor points:

The context of this paper is GCMs, however most of what is discussed is equally relevant to any atmospheric model that uses physical parameterizations, either for processes of diagnostic fields. It may be worth modifying a few things, such as the title and the description of the context (currently framed as for GCMs only), in order to communicate explicitly that this tutorial is applicable to other types of models as well. It would thus appropriately be targeted to a broader audience.

>> Good point. Noted the applicability to other types of models in the introduction in a few places.

Line 29, "… to understand how a diagnostic or parameterization is constructed." It might be useful to provide a bit of background on the various types of parameterizations and diagnostics that are used in atmospheric models. This paper is a tutorial, so the appropriate degree of background is helpful.

>> Added a few more sentences describing different parameterizations and emphasizing their use across scales.

Further, for the rainbow diagnostic itself you use information from the radiation scheme, convection scheme, cloud-fraction scheme, etc., so some description on these parameterizations would help one further understand the rainbow scheme.

>> These parameterizations are described in Section 3.1. We have added a bit more explanation of how each scheme works together for the readers less familiar with atmospheric modeling

Line 5: "it's" should be "its"

>> Changed

In a few places, abbreviations are defined with the expansion capitalized (e.g. lines 1, 72, etc.). I think this is only correct if it is a proper noun. Thus, one should probably write "earth system modes (ESMs)", "general circulation models (GCMs)", etc. for general terms but "Community Earth System Model version 2 (CESM2)" for proper nouns.

>> Corrected (GCM and ESM)

Line 114: Why use max(Ac, Asr) and not Ac+Asr?

>> max(Ac,Asr) represents the total cloud cover (maximum overlap). Noted now in the text.

There are a few "casual" grammatical errors throughout.  E.g., line 147-148, "But there do not… And of course…" (which negate these as sentences).  Just commenting – I realize that this is a somewhat "conversational" article.

>> Clarified these statements and made less conversational.

Regarding lines 306-309, I suppose another approach to use for high-resolution model configurations would be to apply the rainbow diagnostic to an upscaled (coarser) grid, computed from the grid-scale values.  That way you could get fractional cloudiness (at 100 km, e.g.) and stick with the 1D parameterization.

>> Good point!  Noted.

In the second-last paragraph of the Conclusion, you allude to the idea that a rainbow parameterization could have a more practical use as a way to quantitatively assess cloudiness and stratiform rain, which are presumably more important model fields.  It is not entirely clear to me how serious you are about being about the claim that the rainbow diagnostic has an actual practical application for evaluation cloud schemes, etc..  Most of the paper presents this an in illustrative exercise in understanding the challenges related to physical parameterizations.  It might be helpful if this were clarified.

>> Clarified: In particular it does provide a simple illustration/diagnostic of the diurnal cycle of clouds and precipitation. Highlighted now in the conclusions.

Line 89, "This is a strange and silent world…". A few years ago I heard you discussing the question "What would life be like in a GCM [due to the assumed physics]?"  Silence was one aspect, but there were others.  It might be fun and instructive to expand a bit on that sentence in this paper, even if some aspects may not be entirely relevant for rainbows -- though all things radiation, clouds, and precipitation are relevant.

>> That is basically described in this paragraph. To make that clear, the 'strange and silent world' comment is moved to the top of the paragraph and a new summary sentence added.

---

## Author Response (AR2)

Reply to Editor:

I only have a couple of minor points concerning the revised manuscript (version 3):

>> We thank the editor for their comments and detailed suggestions, and have made the corrections noted below.

- l 18 : as requested by Reviewer #2 please change "Earth System Models" to "Earth system models"

>> I have changed  "Earth System Models" to "Earth system models"

- l 53 : change "look at the what" to "look at what"

>> Done: changed "look at the what" to "look at what"

- legend of Figure 1 : should not it be "Figures 3 and 5"?

>> Yes, the caption has been corrected to say "Figures 3 and 5"

- l 300 "increases are large in the sub-tropics and into higher
latitudes": over ocean you mean? This part of the sentence could be
removed as you indicate a similar information in the following sentence.

>> I have removed this part of the sentence as suggested

- l 380 we provide more details

>> Changed: detail → details

- l 457 spherical geometry

>> Corrected: geomerty → geometry

Finally, in his response document, the author indicates several times
"Added ...". It would have been clearer, as done once this response document, to repeat
in this response document what has been added in the manuscript.

>> Sorry about that. I will be more complete next time.